# Mitigating Vaccine Hesitancy and Building Trust to Prevent Future Measles Outbreaks in England

**DOI:** 10.3390/vaccines11020288

**Published:** 2023-01-28

**Authors:** Sarah Thompson, Johanna C. Meyer, Rosemary J. Burnett, Stephen M. Campbell

**Affiliations:** 1School of Health Sciences, University of Manchester, Manchester M13 9PL, UK; 2Department of Public Health Pharmacy and Management, School of Pharmacy, Sefako Makgatho Health Sciences University, Molotlegi Street, Pretoria 0208, South Africa; 3South African Vaccination and Immunisation Centre, Sefako Makgatho Health Sciences University, Molotlegi Street, Pretoria 0208, South Africa; 4Department of Virology, School of Medicine, Sefako Makgatho Health Sciences University, Molotlegi Street, Pretoria 0208, South Africa

**Keywords:** vaccine hesitancy, measles, measles, mumps and rubella vaccine (MMR), 3c model, England, social inequalities, health inequalities

## Abstract

Measles, a highly infectious respiratory viral infection associated with severe morbidity and mortality, is preventable when coverage with the highly effective measles, mumps and rubella vaccine (MMR) is ≥95%. Vaccine hesitancy is responsible for measles outbreaks in countries where measles had previously been eliminated, including in England, and is one of the ten threats to global public health identified by the World Health Organization (WHO). Official administrative 2012–2021 data on measles incidence and MMR coverage in England were reviewed alongside a scoping literature review on factors associated with MMR uptake in England. Whilst measles incidence has reduced significantly since 2012, sporadic measles outbreaks in England have occurred with geographic disparities and variations in MMR coverage. Over the last decade, MMR uptake has fallen across all regions with no area currently reaching the WHO target of 95% coverage of both doses of MMR necessary for herd immunity. Factors associated with MMR coverage overlap with the 3C (convenience, complacency and confidence) model of vaccine hesitancy. The COVID-19 pandemic has reinforced pre-existing vaccine hesitancy. Increasing MMR uptake by reducing vaccine hesitancy requires allocated funding for area-based and targeted domiciliary and community-specific immunisation services and interventions, public health catch-up campaigns and web-based decision aid tools.

## 1. Introduction

### 1.1. Epidemiology of Measles

Measles is caused by the measles virus, which is primarily transmitted via respiratory droplets and aerosols [1,2,3,4]. It is one of the most communicable viral diseases, with a basic case reproduction number (R_0_) of 12–18 [3,5]. It is estimated that 90% of susceptible individuals will become infected with measles after exposure to the virus [6]. Symptoms usually start with fever, cough, coryza and conjunctivitis, followed after a few days by a characteristic rash starting on the face and behind the ears [2]. Measles affects mostly children under the age of five years, with the potential for serious complications including pneumonia, encephalitis, chronic immune suppression and death [1,3,4,6,7].

### 1.2. History of Measles Vaccination Schedules in the UK

Crucially, measles is preventable with vaccination. The first measles vaccine was introduced in England in 1968 as a measles-only vaccine; however, coverage was insufficient to interrupt transmission [1]. In 1988, the measles, mumps and rubella vaccine (MMR) was introduced in England as a one dose schedule at 12–13 months [1]. Although MMR coverage at that time was over 90%, which significantly reduced the transmission and incidence of measles, small outbreaks still occurred. Thus, a two-dose schedule was introduced in 1996, with the second dose being administered at 40 months of age to allow children to receive their second dose before starting school [1]. Children who missed this second dose can also be caught up either as teenagers before they leave school, or as adults [1]. A second dose serves two purposes: the first being a catch-up dose for children who missed their first dose; the second being to increase immunity, as up to 10% of children are not fully immune following their first dose [1]. A recent Cochrane review of 124 studies assessing vaccine effectiveness showed that the first dose of MMR was 95% effective in preventing measles, whereas two doses raised this to 96% [8].

### 1.3. Measles Reporting and MMR Coverage Target for Measles Elimination in the UK

Measles is a notifiable disease in England, meaning any suspected cases must be reported to the local health protection teams, and laboratories must report confirmed cases [9]. Confirmed cases of measles in the UK have only been recorded officially since 1995; however, data for measles notifications and deaths are available from 1940, when statutory reporting of measles was introduced [7,10]. The validity and immediate usefulness of notification data without laboratory verification are questionable, given the likelihood of misidentification and under-reporting [11].

Given the notifiable status of measles, it is a legal requirement to report suspected cases, which are then confirmed using laboratory oral fluid testing. If confirmed as positive for measles, the strain is then further sequenced and entered into the World Health Organization (WHO) global Measles Nucleotide Surveillance system to support the investigation of transmission and sources of infection [12]. However, measles is widely underreported when using this system, particularly in times of sporadic cases and in children younger than 5 years of age [13]. Other countries have indicated that their measles incidence rates are 2 to 36 times higher than reported [14].

Measles elimination is defined as the absence of endemic measles transmission within a defined geographical area for at least 12 months. In addition, elimination can only be verified if a high-quality surveillance system capable of testing at least 80% of suspected measles cases is in place and shows that elimination has been sustained for at least 36 months [4,15,16,17]. Due to the high infectivity of the measles virus, a high level of herd immunity is necessary to interrupt measles transmission. This can be achieved by reaching 95% MMR coverage [18], a target that was set by the WHO in 2001 [19].

### 1.4. Vaccine Hesitancy

Vaccine hesitancy refers to a delay in the acceptance or refusal of vaccines despite the availability of vaccination services [20], and in 2019, it was declared by the WHO as one of the ten leading global threats to public health [21]. Vaccine hesitancy is very complex and influenced by multiple factors grouped into three overarching categories, namely convenience, complacency and confidence, known as the WHO/SAGE 2014 ‘3C’ model of vaccine hesitancy [20,22] (see Figure 1).

Complacency arises when diseases prevented by vaccines no longer occur. Thus, in countries where herd immunity against measles was sustained by 95% MMR coverage, this led to complacency, which in turn led to pockets of low MMR coverage and allowed measles outbreaks to occur [23]. Confidence can be considered in terms of public perceptions about MMR’s safety and effectiveness; public trust in the system that delivers vaccination services; public trust in the motivations of policymakers who include MMR in the National Health Service’s (NHS) childhood vaccination schedule [20]. The UK has been shown to have very high confidence in vaccines generally [24]. However, in relation to MMR specifically, confidence was widely undermined due to the Wakefield 1998 paper, discussed in Section 3.3, and has yet to fully recover [23]. Convenience is measured by the extent to which uptake is affected by physical availability, affordability, accessibility and language and health literacy [20].

### 1.5. Aim of this Review

This review aims to: (a) describe measles incidence and MMR coverage in England over the last decade; (b) identify potential factors associated with MMR uptake in the context of the WHO/SAGE 2014 ‘3C’ model of vaccine hesitancy relating to convenience, complacency and confidence (see Figure 1); (c) identify possible interventions to improve MMR uptake.

The specific focus on measles is because rubella still maintains its status of elimination in the UK, and mumps has yet to be eliminated anywhere in the world [16]. Notably, endemic transmission of rubella is not present in the UK, despite protection being provided from the same vaccination as measles; however, this may be related to rubella’s lower reproduction number of <5 [25] compared to that of measles, which is estimated to be between 12 and 18 [3,5]. Given the complexity of different healthcare service structures across England, Scotland, Wales and Northern Ireland, England has been selected for this review as a specific focus but with wider transferable learning.

## 2. Materials and Methods

Official administrative data on the incidence of measles and MMR coverage in England from 2012 to 2021 were reviewed. Sources included Public Health England (PHE), which was responsible for collecting data on vaccination coverage from Child Health Information Systems (CHIS) from local authorities (LA) or general practices (GP) until it was disbanded in 2021, and its replacement, the UK Health Security Agency (UKHSA) [26]. Quarterly statistics are compiled under the Cover of Vaccination Evaluated Rapidly (COVER) programme and published annually by NHS digital as national statistics [27]. For the purposes of COVER, England is divided into 9 regions: North East; North West; Yorkshire and the Humber; East Midlands; West Midlands; East of England; London; South East; South West [27].

Because a key issue is to ascertain reasons behind any regional variation identified and also whether this is specific to MMR or seen with vaccines generally, data relating to cases were collected from searches of UKHSA, previously PHE, and the Health Protection Agency’s (HPA) website archives [28,29]. Data from the COVER programme were sourced via the digital records of the UKHSA and NHS [27,30]. Data from government sources, the Office for National Statistics (ONS) and the NHS were used to illustrate the burden of measles and MMR uptake.

An adapted scoping literature review was used to consider factors associated with MMR uptake over the last decade in England, including vaccine hesitancy. The search strategy used the MEDLINE, EMBASE and Cochrane Review databases to identify articles reporting on possible factors contributing to the reduction in MMR uptake, with England as the study population (see Table A1 and Figure 2). A digital ‘hand search’ of relevant journals was also conducted to identify articles that were yet to be indexed, ensuring up-to-date research was included and considered. These journals included *British Medical Journal*, *British Journal of General Practice*, *Public Library of Science*, *The Lancet* and *Vaccine*.

Eligibility criteria included: published after 1990; reference to UK population with only studies involving England; focus on MMR uptake; published in English. Exclusions included news or comment articles, outbreak studies and research focussing on mumps or rubella. Identified articles were appraised using the Critical Appraisal Skills Programme and Critical Appraisal Tool to evaluate the strength of evidence and, thus, critique the confidence in the conclusions drawn [31].

The results of the literature review of factors associated with MMR uptake were assessed in relation to the SAGE/WHO 2014 ‘3C’ model of vaccine hesitancy [20].

## 3. Results

### 3.1. Measles Cases in England from 2012 to 2021

The year 2012 was selected as the starting point for this review because after almost being eliminated in England in the early 2000s, endemic measles transmission was re-established by 2006, and measles incidence continued to rise until peaking in 2012. This increase in measles incidence follows the legacy of children in the late 1990s not receiving MMR following publication of a discredited (and subsequently retracted) 1998 paper in The Lancet by Wakefield and colleagues, which spuriously suggested a link between MMR and autism [33,34], as discussed in Section 3.3.

As illustrated in Figure 3, a steady reduction in cases until 2015 was followed by a steep increase in 2016, which peaked in 2018. However, these cases did not result from endemic measles transmission but were imported from ongoing outbreaks across mainland Europe [15,35]. Thus, because endemic transmission in England and the rest of the UK had been interrupted since 2014, in 2017, measles was declared eliminated in the UK by the WHO Regional Verification Commission (RVC) for Measles and Rubella Elimination [15]. However, by 2019, endemic measles transmission was again established in England, causing the UK to lose its measles elimination status [16]. The numbers of confirmed cases in 2020 and 2021 were significantly lower, which is likely due to the impact of lockdowns and health-seeking behaviour during the coronavirus disease 2019 (COVID-19) pandemic [36].

Contrary to the WHO elimination indicator of at least 80% of suspected cases being tested for confirmation of infection [37], in the quarter of July to September 2021, only 65% of suspected cases were tested in England [36]. While it has been suggested that these confirmed case numbers were low because laboratories received a lower number of specimens and were overwhelmed by COVID-19 testing during this period [38,39], this measure of confirmed cases was consistent over the last 5 years, suggesting that COVID-19 had a minimal impact on the ability of laboratories to provide testing for measles infection [40]. However, this low rate of confirmation of suspected measles infections suggests that some cases may not be included in the dataset.

Table 1 shows the number of confirmed cases of measles in England by region for the period 2012 to 2020. While the crude number of measles cases across the regions of England shows significant disparities, this may reflect differences in population size.

To consider the true burden of measles, the figures must be adjusted for regional population estimates during the periods of 2012–2020; thus, they are presented here as a rate of cases per 100,000 population (see Table 2). Currently, the highest population density is seen in the South East, and the lowest is in the North East [42]. While there has been some variation across the last decade, London has witnessed the highest incidence of measles since 2014.

### 3.2. MMR Coverage in England from 2012 to 2021

#### 3.2.1. National MMR Coverage

The official statistics in England measure the percentage of children who have had their first dose of MMR (MMR1) by their second birthday and their second dose (MMR2) by their fifth birthday, aligning with NHS childhood vaccination guidance [27]. The most recent COVER statistics demonstrate that the uptake of both MMR doses declined to the lowest in the last decade in England (Figure 4). MMR coverage increased from 2013 but began to decline after 2017. No regions in England reached the WHO target of 95% coverage of both doses of MMR.

In 2013, an MMR catch-up campaign was launched by PHE, NHS and the Department of Health in order to increase MMR coverage. The aim of the campaign was to ensure at least 95% of previously unvaccinated students aged 10 to 16 years had received at least one dose of MMR [43]. The results suggested that 11% of the target population were reached through the catch-up campaign at mid-point [43].

The target of 95% MMR1 coverage was achieved for the first time in the UK in 2016 [1,18]. MMR1 and MMR2 coverage in the UK remained consistently below the 95% coverage target over the last decade and is declining, causing a rise in cases [27,29]. The most recent quarterly data report 88.6% and 85.5% for MMR1 and MMR2 coverage in England, respectively [30], with significant variation across UK countries and regions within England [22,30].

#### 3.2.2. Regional MMR Coverage in England

In addition to national statistics, COVER provides regional MMR coverage (percentages) for geographical comparison. Table 3 shows MMR1 coverage in England by region from 2012 to 2021, with the lowest coverage shown in red and the highest in blue.

Reflecting the dataset from confirmed cases (Table 2), London also consistently showed the lowest MMR coverage (Table 3). Coverage in the North East was consistently high, often reaching the WHO target of 95% [44].

Table 4 shows that continuing to age 5, London maintained the lowest MMR coverage in England, with North East at the peak. Multiple regions reached the WHO target for the first dose of MMR by 5 years in 2021; however, all regions fell short with both doses. These regional discrepancies were reflected in the uptake of the 5-in-1 (against diphtheria, tetanus, pertussis, polio and Haemophilus influenzae type b) vaccine, of which the North East had the highest uptake in 2021 at 97.2%, and London was the only region that did not reach the WHO 95% coverage target [27]. This suggests that regional differences are not specific to just MMR.

#### 3.2.3. MMR Coverage in England by Socioeconomic Status

Similar to geographical location, socioeconomic status also had an impact on MMR uptake (Figure 5). The “English indices of deprivation 2019”, which divides England into ten deciles statistically ranked from least to most deprived, were used. Deprivation is calculated based on income; employment; health deprivation and disability; education and skills training; crime; barriers to housing and services; living environment [45].

Compared to the national average MMR coverage for 2021, the most deprived decile showed lower coverage across all three measures. London previously showed a high deprivation index, which may have contributed to the maintenance of its low coverage [43]. However, some districts in London saw a relative decrease in their levels of deprivation between 2015 and 2019, the period when the indices were last reviewed [46]. Given that their measles incidence and MMR coverage was continually problematic, yet their deprivation level was improving, this suggests that there were other factors at play. Additionally, despite having the highest MMR coverage, the North East scored the highest average level of deprivation of all nine regions in 2019, with 49% of the Middlesborough neighbourhoods being in the top 10% of most deprived neighbourhoods in England, representing the highest proportion of deprivation in the country [46]. These data indicate that although socioeconomic status may be involved, its impact on MMR coverage is more complex. Furthermore, regional differences are not specific to MMR, and the reasons for general vaccine hesitancy in these areas must be explored and addressed appropriately.

### 3.3. Literature Review: Potential Factors Associated with MMR Uptake

The adapted literature review identified several factors associated with MMR uptake and vaccine hesitancy in England (Table 5). The colours in Table 5 align to Figure 1 and are expanded upon in reference to the 3Cs framework [20].

#### 3.3.1. Convenience 

##### Accessibility 

The lack of opportunities to vaccinate children conveniently outside of the standard NHS system negatively impacted parents who chose to vaccinate their children, but were unable to attend the appointment due to illness and not being able to rebook [24,47], or because they worked during the same hours that a GP practice was open [24]. The lack of convenient access has been referenced in previous outbreak research, including the 2013 outbreak within the orthodox Jewish population of London. There was speculation that there may be religious or philosophical objections to MMR; however, the main obstacle cited was the inconvenience of taking children across the city on public transport [48]. While lack of access is referenced as a factor for all communities [49] it is strongly referenced as a factor for Roma communities in the UK [22], and membership of traveller communities is not currently recorded or monitored by the NHS [15], making assessing their immunisation uptake and developing services to meet their communities’ health needs challenging.

Those from more deprived areas were found to be significantly less likely to have received a full course of COVID-19 vaccines, suggesting that many of the contextual MMR factors are widely applicable to vaccines in general [50]. For example, those living in larger households were less likely to have received a COVID-19 vaccination and boosters, which is also seen with MMR and is directly related to socioeconomic status [50].

Over the course of the pandemic, most countries have shown increasing vaccine acceptance for COVID-19 [51]. The strategies in place for limiting COVID-19 have also mitigated the spread of measles; thus, lower MMR coverage has not yet translated to rising case statistics [52,53].

##### Population Mobility

It is probable that COVID-19 has had an influence on MMR uptake in England [52,54], which was found to be lower during the initial lockdown period although the opposite was found in Scotland, where there were active measures to promote immunisation at local and national levels [54]. Furthermore, health visitors, who are traditionally the first contact of new parents to discuss and encourage vaccinations for their children, were redeployed during the pandemic to support other services [55].

##### Information on Service Availability

Other countries have also cited parents being unaware that a second dose of MMR is required for optimal immunity; if this were also the case in England, then it would provide some explanation for the lower percentage of children vaccinated with MMR2 compared with MMR1 [27,56,57].

Guidelines were published on delivering MMR in primary care settings throughout the COVID-19 pandemic [58]. However, 1 in 10 parents who had not vaccinated their child against measles said they were unaware the NHS was still offering appointments [59].

##### Affordability and Funding

There has been inequitable funding for GP practices across England [60,61], with practices serving deprived communities and transient populations being relatively underfunded in comparison with practices serving more affluent communities [62]. This negatively impacted MMR uptake in these deprived communities [22,61]. Additionally, London had a lower ratio of GPs to patients than the North East [63], which may indicate that access to GP services has an impact on vaccine coverage.

Underserved and hard-to-reach communities had lower routine vaccination coverage. This includes travellers, migrants and looked-after children (i.e., a child cared for by their LA for more than 24 h) [22].

##### Inequalities and Sub-Populations

Certain populations and sub-groups in England were more likely to have lower vaccination uptake, which may also be contributing to the geographical variations and inequalities. MMR coverage was much lower in urban populations than rural areas [22,64].

Children with learning disabilities were much less likely to be fully vaccinated with MMR than their peers [65,66].

Social and health inequalities (i.e., systematic differences in health status or the distribution of health resources between population groups that are unfair or preventable [22]), including language barriers, low literacy, discrimination, poor school attendance, poverty and inadequate housing, were identified as barriers to MMR uptake across different communities [15]. These inequalities resonate with the negative social determinants of health (SDH), which must be addressed to ensure health for all. The SDH are defined by the WHO as “the conditions in which people are born, grow, work, live, and age, and the wider set of forces and systems shaping the conditions of daily life” [67,68,69].

Religious or cultural beliefs prohibiting the use of animal products and animal testing in creating and manufacturing vaccines is cited as a reason for not vaccinating with MMR [70,71]. Measles is grown in chick embryo cells, and one of the two available brands of MMR contains porcine gelatine [72].

Inequalities in England were found to have an impact on most parts of life, including life expectancy, educational attainment and crime [73]. Disparities in service provision for areas of higher inequality were widely cited as negatively impacting vaccination rates, both MMR-specific and in general [74,75,76,77].

This literature review shows considerable focus on the low MMR uptake in London, but less attention has been paid to the higher uptake in the North East, which could identify key support structures and be used to create a framework to apply to other regions [48,78].

A notable demographic difference between London and the North East are the differences in ethnicity, given that London has the lowest percentage of white British people, whereas the North East has the highest [79], and there are lower levels of immigration in the North East than other regions [80].

#### 3.3.2. Confidence

##### Vaccine Safety and Effectiveness

A recurring misperception regarding MMR is the belief by some parents that single vaccines are safer and more effective than combined vaccines, and that giving multiple vaccinations can overload the immune system, leading to safety concerns [24,70,81]. Some parents believed their child’s immune system would not be able to cope with multiple immunisations, reflecting the lack of trust in combined vaccines [24,70].

Many parents were unaware of the protective qualities of MMR [69].

Although the work is now utterly refuted, Wakefield’s 1998 paper on autism is still referenced by concerned parents, and it influences confidence in and perceptions of MMR [82]. A study conducted during an outbreak in Merseyside found over half of respondents referenced autism directly as a concern [24]. Previous work has also concluded that parents perceive MMR as more dangerous than other childhood vaccinations [83], highlighting the need for behavioural and educational public health intervention.

##### Healthcare Professionals

Trust in and support from healthcare professionals are both important in increasing confidence in, and uptake of, MMR. Parents have widely stated that they value the input of healthcare professionals as a trusted source of vaccination information [83,84]. Parents desire open discussion with healthcare professionals, to weigh their risks, benefits and options in an unbiased manner to help them make an informed decision [23,70]. Many parents, both those who accept vaccination and those who do not, have reported perceived unwelcome pressure from healthcare professionals to vaccinate, making their decision more stressful and difficult [66,84,85]. Research in Italy has previously identified that receiving satisfactory information from healthcare professionals was significantly associated with acceptance of, and confidence in, MMR, suggesting that this could be usefully applied to England [61,86].

##### Trusted Information

In addition to issues around mis- and disinformation discussed in other sections, there are specific topics where trusted information was reported as absent. For example, there is confusion relating to breastfeeding and vaccinations, resulting in the misconception that because breast milk transfers immunity, children do not need to be vaccinated until they are weaned [66,87]. The NHS childhood vaccination schedule is based on evidence that passive immunity against measles wanes before a child’s first birthday; however, information on this is not readily available [87,88].

##### Mis- and Disinformation (Media and Autism)

In 1998, Andrew Wakefield and colleagues published a now-infamous report, which suggested that MMR may predispose infants to gastrointestinal disease and developmental regression or autism [34]. In spite of its small sample size and tenuous conclusions, it received significant publicity and negatively impacted the confidence in and uptake of MMR in the UK [81,82,89]. It was accompanied by misinformation in the media and public health scares, heavily publicising headlines and opinion articles relating to parents’ stories of their children and MMR. For example, areas served by the South Wales Evening Post newspaper saw a statistically significant drop in MMR uptake compared to the local average following their three-month campaign questioning MMR, and the media had a negative impact on its perceived acceptability in the communities that it reached [90].

The epidemiological studies that followed refuted these claims, and the Wakefield paper was later retracted by the authors and publisher following revelations of fraud and failure to disclose financial interests [91]. A steep rise in measles cases was observed in 2013, which was attributed to unprotected adolescents who missed out on vaccination between 1998 and 2003 during the disproven autism scare [10]. In recent years, it was also recognised that vaccine deniers gained significant traction on social media platforms, likely contributing to the decline in MMR uptake [92,93]. This phenomenon is not specific to England, nor to MMR, and is reflected in vaccine coverage statistics across the world [94,95]. A survey by the Royal Society for Public Health found that 50% of British parents of children younger than 5 years of age have regularly encountered negative messages about vaccination on social media [92].

##### Personal Experience

Some studies have reported that parents having contracted measles as a child was not a good indicator for accepting MMR for their children [70], whereas others reported that parents who witnessed the severe outcomes of measles in their immediate family were more likely to accept MMR for their children [96]. In addition, parents who experienced autism within their families, or knew children with autism, were less likely to accept MMR for their children [23]. Furthermore, some parents felt the vaccine was unnecessary and instead turned to nutrition, fitness or homeopathic remedies [23,24,70].

#### 3.3.3. Complacency

##### Awareness of Severity of the Disease

A large volume of literature refers to parental attitudes in relation to MMR coverage [81,96]. However, the severity of measles is not widely understood by parents. A survey of English parents of children aged under 5 years reported that over 60% of respondents were unaware that measles can be fatal, and almost 50% were not aware that it can lead to serious complications [59]. Vaccine-hesitant parents in the UK [23] and across Europe viewed measles as low-risk [97], with almost 1 in 5 parents who had not had their child vaccinated for MMR not aware that measles is a threat in the UK [59]. This low perception of disease severity and risk is associated with complacency, leading to lower uptake [24]. This is demonstrated by an increase in uptake during times of measles outbreaks [98].

##### Perceived Risk of Disease

Generally, vaccination coverage is often higher during times when disease is highly visible [99]. For example, the WHO declared COVID-19 a public health emergency of international concern in January 2020 and a pandemic in March 2020, when all occupied continents were affected [100]. The R_0_ for COVID-19 was estimated at 2.2 in Western Europe, significantly lower than that of measles [5,100].

### 3.4. Literature Review: Possible Interventions to Improve MMR Uptake

This paper has shown that measles is endemic in England, and that MMR coverage is not at sufficient rates to provide herd immunity in England. The potential factors associated with reduced MMR uptake and hesitancy are myriad, with multiple interventions needed at macro, meso and micro levels to improve uptake. The WHO recommends periodic intensification of routine immunisation, which describes a spectrum of time-limited, intermittent activities and campaigns to target under-vaccinated populations and raise awareness [101]. Table 6 shows the key issues, challenges, demographic and economic considerations associated with low MMR uptake that were identified in previous sections together with recommendations to address these barriers. While the recommendations relate to improving MMR uptake in England, they have transferable learning globally for all vaccination programmes.

Interventions are needed to reduce vaccine hesitancy and reduce avoidable health care costs necessitated by mortality and morbidity resulting from not receiving MMR. A large amount of the literature base in England focuses only on call–recall systems [102,103,104,105], where automatic communications are sent to parents and guardians for children who are due (call) or overdue (recall) for MMR vaccination. However, there is a need to consider convenient access, including affordability in countries where vaccination is not free, as well as service provision, including available workforce [106]. In addition, the attitudes and behaviours of parents and guardians must be considered, as behaviour change is fundamental to tackling social challenges [107]. There are a wide range of behaviour change techniques and taxonomies available [108].
vaccines-11-00288-t006_Table 6Table 6Barriers and recommendations.BarriersRecommendationsGeneral vaccine hesitancy Interventions targeted at barriers of convenience, complacency and confidence [23,44]Behavioural change interventions [107,108]Lack of perceived trusted information [24,59,70]; concerns over combined vaccines [24,70,81]; concerns over side effects [44,65,82]Annual countrywide measles and MMR awareness programmeWeb-based decision aid tool providing clear, unbiased and evidence-based information on measles and MMROfficial fact-checking for immunisation posts on social media and signposting to valid sources.Lack of awareness of measles severity and availability of MMR vaccination programme [23,24,59]Annual in-school education campaignsRadio and TV advertisement messagesHighlight routine availability of MMR not containing porcine gelatineInequalities in service provision [22,65,73,74,75,76,77,103]Annual in-school MMR catch-up and education campaignsOut-of-workday hours vaccination availabilityFocus on populations most likely not to vaccinate i.e., living in larger households, traveller communities, migrants, looked-after children [23] and children with learning disabilities [65]Foster public health programmes and ethos as in the North East of England [44]Community-specific needs are not addressed [23,24,71,78]Area-based community and domiciliary immunisation teamsHighlight routine availability of MMR not containing porcine gelatine [72]Inadequate immunisation training [9]Vaccine-specific training for healthcare professionals [51,77]Effective communication and cultural awarenessMMR uptake awareness to encourage opportunistic vaccine checks and reduce missed opportunities [24,109]Lack of advanced information technology for surveillance [13,40]Use novel methods of COVID-19 contact tracing to enhance measles surveillanceLack of funding [9]Allocated funding for community and domiciliary immunisation teamsUse high-visibility campaigns to encourage funding [101]Vaccine availability and affordability [20,110]This has historically not been an issue in England [111]Novel vaccine delivery innovations including microarray patches, slow-release preparations and enhanced thermostability [112]

#### 3.4.1. Convenience

Implementing strategies at individual and community levels to improve MMR coverage in England will likely reduce the risk of outbreaks, reduce health inequalities and provide applications to other vaccination programmes [20]. This resonates with wider quality improvement literature that emphasises systemic approaches with interventions at micro, meso and macro levels that reinforce each other [113].

Whilst the majority of intervention literature focuses on parental decision-making, conducting catch-up campaigns amongst different age groups can help prevent outbreaks as well as educate and encourage the child or adolescent instead of their parent [43,96,110]. Studies have found that school-based campaigns to catch-up on MMR for adolescents are particularly effective in comparison with signposting to GPs [110]. There is a need for greater focus on adolescent and adult immunisation against measles. While there is some evidence related to mumps and unvaccinated university students, in addition to MMR catch-up campaigns, research into acceptability and methods of targeting is limited [43,114].

The same applies to the ‘convenience’ issue: addressing the fact that many parents work during the same hours that a GP practice is open [24]. A wider choice of clinics and venues, including places within the community, that have extended opening hours or weekend availability would increase MMR uptake [24]. However, this would require workforce planning and funding. For example, the feasibility of implementing parent-centred, multi-component interventions in different communities across England would require additional staff to cover more appointments and potentially longer consultations [61].

A key component of outbreak studies is community outreach and engagement, including domiciliary vaccination and community clinics [115]. The National Institute for Health and Care Excellence (NICE) supports the use of a domiciliary immunisation service as a cost-effective use of NHS resources in increasing MMR uptake and reaching vulnerable children [116]. Although there is evidence of the success of these interventions, they are community-specific and often short-lived due to lack of funding and insufficient programme evaluation [15].

Despite this lack of funding, interventions to increase MMR uptake have been demonstrated as a cost-effective alternative to outbreak management: the estimated cost of the Merseyside outbreak in 2012 was £4.4 m, including NHS treatment, public health costs and societal productivity losses. In comparison, over the previous five years, a further 11,793 MMR administrations would have been needed in Cheshire and Merseyside to achieve herd immunity at an estimated cost of £182,909, only 4% of the total cost of the measles outbreak [117]. Furthermore, a review of international studies suggests that preventative public health intervention has a long-term social return on investment, averaging £14 of social benefit for every £1 spent [118].

The presence of animal products and animal use in vaccines have been recognised as disincentives in uptake [70,71]. This is often related to lifestyle factors, including veganism and religion [71,119]. The presence of porcine gelatine in particular is often cited for vaccine refusal, indicating the need for Halal vaccines [120]. However, there are two types of MMR widely available in the UK, and one does not contain porcine gelatine; therefore, more must be done to highlight the availability of this type of MMR [10,118].

The COVID-19 pandemic offers a platform for accelerating progress towards the elimination of measles. Methods of contact tracing have evolved and could be repurposed to enhance measles surveillance [121].

#### 3.4.2. Confidence

The continued concerns about side effects and autism is a remnant of the discredited Wakefield article, which continues to impact MMR uptake over 20 years later [34,82]. The fact that it is still referenced when explaining lack of confidence in vaccines despite being widely disproven speaks to the need for behavioural and educational interventions. Vaccine deniers have gained significant traction on social media platforms, leading to the spread of misinformation and bias [93,94]. Global advocacy is needed for standardising online fact-checking for immunisation posts on social media. Fact-checking posts have become increasingly common, particularly throughout the COVID-19 pandemic, and have demonstrated their necessity in increasing vaccine confidence and positive attitudes [122].

Strategies to address misinformation and build vaccine confidence must address behavioural attitudes that underpin vaccine hesitancy that are embedded in societies. In 1869, the ‘Leicester Method’ of quarantine was used to counter cholera and was accompanied by a contemporary anti-vaccination movement, showing that the human psychology of vaccine hesitancy is deeply ingrained, and without addressing the root causes, anti-vaccination movements will continue into the future [123,124]. Interventions must focus on the lack of fact-checked information and awareness; dis- and misinformation about autism, side effects and vaccines generally; personal experience; access; population mobility; media; inequalities; combined vaccines; funding; healthcare professionals and workforce.

A lack of perceived trusted information is key in the decision not to vaccinate both oneself and one’s children [24,59,70], requiring targeted interventions tailored for specific communities [125]. The COVID-19 pandemic showed that a relatively large proportion of the global population does not trust vaccination [126]. The levels of lack of trust in vaccination may differ between different sub-populations within countries. For example, a study on COVID-19 vaccine hesitancy in the UK reported that 29.2% of Black participants did not trust vaccines, whereas 35.4% of Pakistani and Bangladeshi participants were concerned about vaccine side effects. In contrast, only 5.7% and 8.6% of white participants did not trust vaccines or had concerns about side effects, respectively. These findings led the authors to conclude that “urgent action to address hesitancy is needed for some but not all ethnic minority groups” [64], reinforcing the need for tailored approaches for different ethnic minority groups [64] and those from lower socioeconomic backgrounds [125]. Similarly, whereas nearly a quarter of UK health workers were reluctant to receive regular COVID-19 vaccination, Black participants were statistically significantly more reluctant to receive vaccination than white participants [127]. In addition, as a result of the Ukraine war and tensions in the Middle East, refugees to the UK, predominantly women and children, may have low levels of MMR coverage [128]. Each population sub-group with lower vaccination undermines the WHO-recommended vaccine coverage of 95% or higher each year to achieve and maintain herd immunity and protect the population. It is imperative to focus not only on populations as a whole but also those groups that are being most impacted by these factors. However, this would require financial resources, human resources and workforce planning.

Interventions to raise healthcare professionals’ awareness of MMR uptake has also been demonstrated as useful [109]. Healthcare professionals must communicate effectively, as they care for increasingly diverse communities with specific needs, varied language abilities and literacy skills [15]. However, nurses and healthcare visitors, who are nurses and midwives working with families and focussing on the health of pre-school children in the UK, consistently report that they do not feel well-equipped to deliver vaccination education as a part of their role [77]. Healthcare professionals should be equipped with the necessary skills to be able to tailor their communication in a non-judgmental, individualistic approach to maintain or establish trust. Educational interventions should therefore include sensitivity and cultural training [23,70].

Multi-channel and media approaches are required, including radio, out-of-home, public relations (PR) and partnerships to reach out to parents and carers of 0–5 year olds to encourage MMR uptake [129]. In addition, to have continued impact, it is recommended that these campaigns should be implemented on an annual or semi-annual basis [101]. An alternative to face-to-face interactions is a web-based decision aid tool, which can increase MMR knowledge and support vaccination decision-making [130]. The provision of tailored and targeted factual information reduces complacency and increases confidence in MMR [83,92].

#### 3.4.3. Complacency

The COVID-19 vaccine has shown itself to be useful in protecting against severe disease, although less effective at preventing transmission. Although this is not the case with MMR, it may have affected parental perceptions of routine childhood vaccinations, and it warrants further investigation [121]. To combat this, it is essential that future vaccination communication highlights that not all vaccines have the same method of action, nor do all vaccination programmes have the same aims [121].

In order to combat complacency, more could be done to highlight awareness of the severity of measles [24], such as GP practice-based posters to display information on the complications of measles [109]. However, this type of intervention fails to address hard-to-reach or seldom-heard populations, including those not registered to a GP, and may also present a language barrier; hence, it is a factor of convenience as well. The WHO recommends using visual communication to overcome this and ensure health literacy is achieved [131], and for communication to be available in multiple languages and in Braille. However, an informative leaflet alone does not significantly improve vaccine uptake, although face-to-face interactions [77] and parent-centred, multi-component interventions including balanced information, group discussion and coaching exercises on informed parental decision-making for MMR both increased knowledge and vaccination rates [132].

## 4. Conclusions

This paper has shown that measles is endemic in England, with outbreaks being driven by low MMR coverage. The incidence of measles across England and its regions shows geographical variation—for example, between London and the North East—and research is needed to explain any local and behavioural reasons behind the higher MMR uptake in the North East.

While socioeconomic status and wider social determinants of health and ethnicity have roles in MMR uptake, with lower coverage in the most deprived areas, those from lower socioeconomic backgrounds and among some ethnic groups for both MMR [22,75] and COVID-19 vaccines [64,125], it is a multifaceted issue highlighted by the use of the 3Cs model of vaccine hesitancy: confidence, convenience, and complacency. There is no question that the discredited article by Wakefield and colleagues reduced uptake and confidence in vaccines (generally and MMR), with a large volume of literature focusing on parental attitudes and the aftermath of the Wakefield controversy. Vaccine hesitancy impacts MMR uptake, and it is particularly influenced by side effect and autism fears. General vaccine hesitancy remains in some groups profiled as more influenced by social media. Importantly, regional disparities are consistent across vaccination types. Moreover, there is a need for tailored interventions based on an understanding of, and motivations for, vaccine hesitancy generally and for different types of vaccinations [125].

“Unlike other medicines, vaccines work at both the individual and community level” [20]. It is imperative to place the issues and challenges surrounding vaccines within the wider issues of a global and national healthcare system. The importance of vaccine hesitancy cannot be overstated. However, approximately half of the world’s population is unable to access essential medicines. The same applies to vaccines as preventative medicine. Even in England, a high-income country, poor access and inequalities in service provision contribute to geographical differences in MMR coverage.

“Achieving universal health coverage and equity in public health depends on access to essential, high-quality and affordable health related technologies for all” [133]. Vaccines must be seen and prioritised as essential medicines, and they must be conveniently provided for timely access. Lack of vaccination is an avoidable harm [134]. However, people can only choose to be vaccinated or have their children vaccinated if they can gain convenient access physically to affordable vaccination delivered in a health facility by an appropriate healthcare worker. This requires system and workforce planning. The WHO, the World Bank Group and the Organisation for Economic and Co-operation Development (OECD) reasserted in 2018 with “The Sustainable Development Goals” a global commitment to achieve universal health coverage by 2030. This means that “all people and communities, everywhere in the world, should have access to the high-quality health services” [106]. They identified five foundational elements critical to delivering quality healthcare services, those being healthcare workers; healthcare facilities; medicines, devices and other technologies; information systems; financing [107]. Finally, the COVID-19 pandemic has shown what can be done by utilising accurate information systems for population vaccination. However, “most OECD health systems invest only 2–4% of total health expenditure in information systems. In most low- and middle-income countries, the figure is less than 1%” [107].

## Figures and Tables

**Figure 1 vaccines-11-00288-f001:**
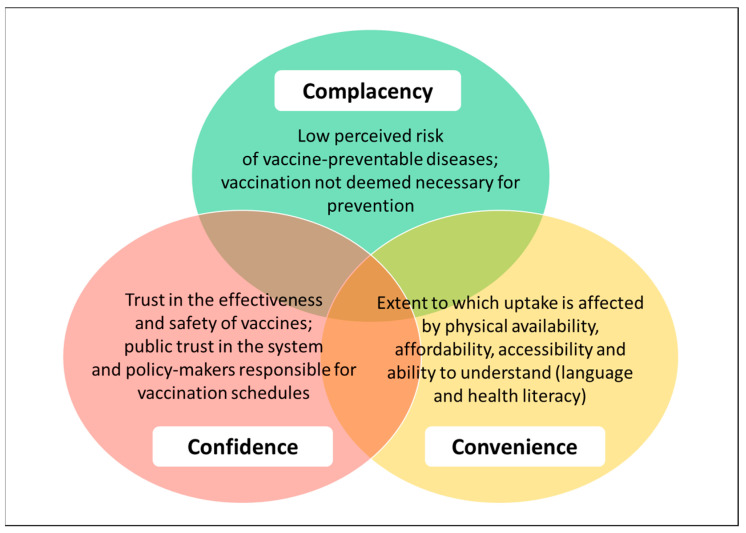
The 3Cs of vaccine hesitancy. Source: adapted from [20].

**Figure 2 vaccines-11-00288-f002:**
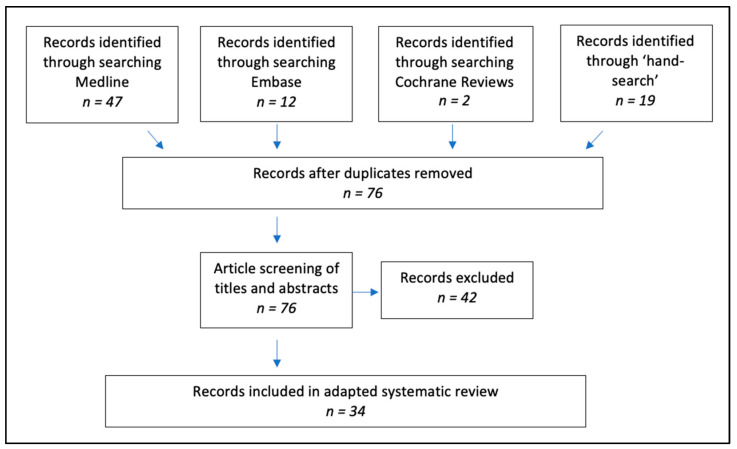
Search strategy for literature answering the research question, “Why has MMR uptake in England reduced over the last decade?”. Source: adapted from [32].

**Figure 3 vaccines-11-00288-f003:**
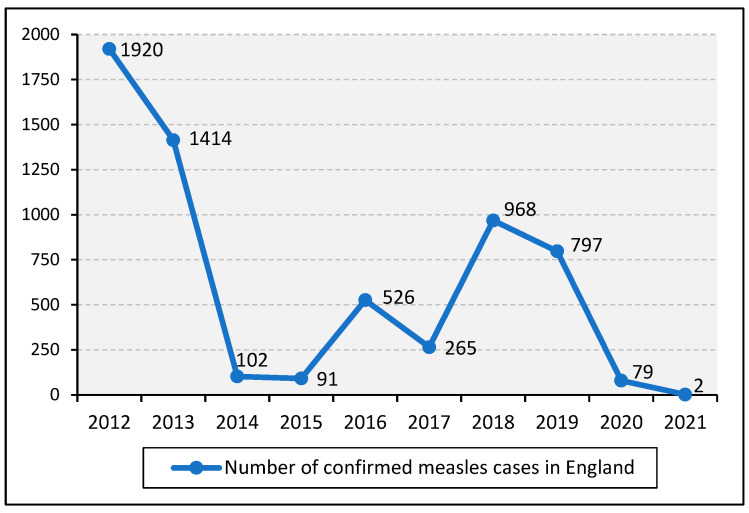
Number of confirmed cases of measles in England from 2012 to 2021. Source: [29].

**Figure 4 vaccines-11-00288-f004:**
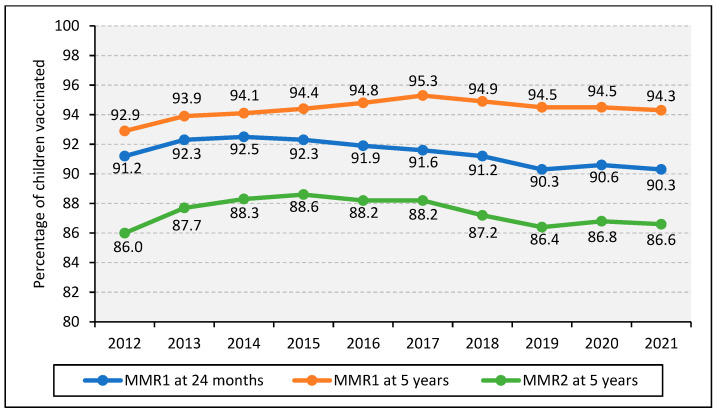
Percentage of children vaccinated with MMR in England from 2012 to 2021. Source: [27].

**Figure 5 vaccines-11-00288-f005:**
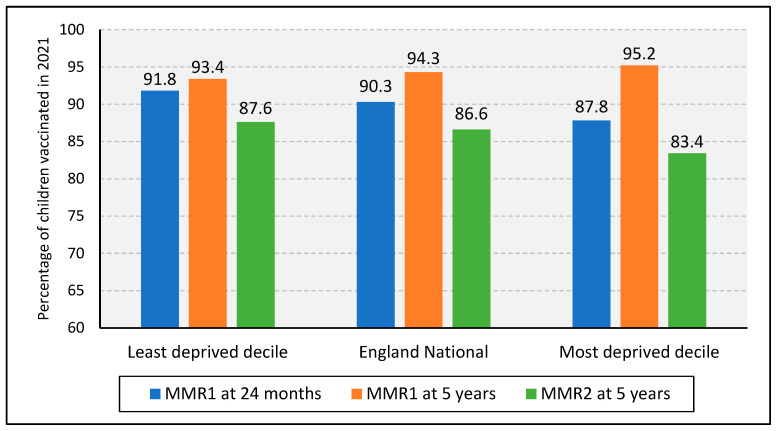
MMR coverage by socioeconomic status in 2021. Source: [45,46].

**Table 1 vaccines-11-00288-t001:** Number of confirmed cases of measles in England by region from 2012 to 2020.

Number of Confirmed Cases of Measles									
Region	North East	North West	Yorkshire and the Humber	East Midlands	West Midlands	East of England	London	South East	South West
**2012**	50	881	152	51	122	49	135	383	97
**2013**	379	381	88	49	64	76	189	26	162
**2014**	6	7	2	4	7	10	51	12	3
**2015**	2	2	2	0	2	1	65	4	13
**2016**	1	3	12	4	8	24	318	60	96
**2017**	2	42	40	1	38	14	68	23	37
**2018**	14	33	84	13	81	33	389	181	140
**2019**	3	51	28	48	14	102	461	68	22
**2020**	0	9	0	4	0	5	39	19	2

Source: [41].

**Table 2 vaccines-11-00288-t002:** Rate of measles infection in England per 100,000 population by region from 2012 to 2020.

Rate of Measles Infection per 100,000 Population									
Region	North East	North West	Yorkshire and the Humber	East Midlands	West Midlands	East of England	London	South East	South West
**2012**	1.92	12.44	2.86	1.12	2.16	0.83	1.62	4.39	1.82
**2013**	14.52	5.36	1.65	1.07	1.13	1.28	2.25	0.30	3.01
**2014**	0.23	0.10	0.04	0.09	0.12	0.17	0.60	0.14	0.06
**2015**	0.08	0.03	0.04	0.00	0.03	0.02	0.75	0.04	0.24
**2016**	0.04	0.04	0.22	0.08	0.14	0.39	3.63	0.66	1.74
**2017**	0.08	0.58	0.73	0.02	0.65	0.23	0.77	0.25	0.67
**2018**	0.53	0.45	1.53	0.27	1.37	0.53	4.37	1.98	2.50
**2019**	0.11	0.69	0.51	0.99	0.24	1.64	5.14	0.74	0.39
**2020**	0.00	0.12	0.00	0.08	0.00	0.08	0.43	0.21	0.04

Source: [41,42].

**Table 3 vaccines-11-00288-t003:** Percentage of children vaccinated with MMR by their second birthday in England by region from 2012 to 2021.

Percentage of Children Vaccinated with MMR by Their Second Birthday									
Region	North East	North West	Yorkshire and the Humber	East Midlands	West Midlands	East of England	London	South East	South West
**2012**	93.0	**93.4**	93.1	92.9	92.0	91.8	**86.1**	92.1	91.7
**2013**	94.1	**94.9**	94.2	94.0	92.7	92.8	**87.1**	92.6	93.5
**2014**	**95.5**	94.8	94.7	94.9	93.6	93.7	**87.5**	91.8	94.2
**2015**	**95.2**	94.0	94.3	94.3	93.5	93.9	**87.3**	91.1	93.7
**2016**	**95.0**	92.9	94.0	94.1	93.1	93.5	**86.4**	91.9	92.9
**2017**	**94.9**	93.6	94.0	93.6	93.2	93.8	**85.1**	90.8	93.1
**2018**	**94.5**	92.9	93.3	93.1	91.2	92.4	**85.1**	91.5	93.3
**2019**	**94.5**	92.4	92.8	92.0	90.6	91.3	**83.0**	91.4	93.0
**2020**	**95.1**	91.9	92.6	92.3	90.3	91.7	**83.6**	92.7	93.3
**2021**	**95.3**	91.0	92.3	92.4	90.2	92.6	**82.4**	92.3	93.3

Source: [27].

**Table 4 vaccines-11-00288-t004:** Percentage of children vaccinated with MMR by their fifth birthday in England by region in 2021.

Percentage of Children Vaccinated with MMR by Their Fifth Birthday									
Region	North East	North West	Yorkshire and the Humber	East Midlands	West Midlands	East of England	London	South East	South West
**MMR1**	**97.0**	95.2	95.8	95.7	94.4	95.5	**88.8**	95.3	96.0
**MMR2**	**92.5**	87.4	90.0	89.0	85.6	90.4	**75.1**	89.5	91.2

Source: [27].

**Table 5 vaccines-11-00288-t005:** Key factors identified from a review of MMR literature.

Factors	Complacency	Confidence	Convenience
Accessibility			**X**
Awareness of disease severity	**X**		
Affordability and funding			**X**
Healthcare professionals		**X**	
Inequalities and sub-populations			**X**
Information on service availability			**X**
Trusted information		**X**	
Mis- and disinformation (autism)		**X**	
Perceived risk of disease	**X**		
Personal experience		**X**	
Population mobility			**X**
Vaccine safety and effectiveness		**X**	

## Data Availability

Additional data are available on reasonable request from the corresponding author. However, all informational sources and papers have been extensively referenced.

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
