# Peer review of "Mitigating Vaccine Hesitancy and Building Trust to Prevent Future Measles Outbreaks in England"

_vaccines, 2023, doi:10.3390/vaccines11020288_

Round 1

Reviewer 1 Report

Thank you very much for sharing this with me. This is an excellent article. I have several minor comments.

1. I recommend the authors to expand their discussion on the role of race, ethnicity, and immigration on vaccine hesitation. 

2. It may be useful to mention the social determinants of health framework. There a range of intervening factors discussed such as health care and misinformation. It may be important to clearly discuss which groups are being most impacted by these factors.

3. Please include the section that provides policy recommendations based on your study findings.

Author Response

Please see the attachment: Response to Reviewer 1 MMR paper_Thompson_20 Jan 2023 final

Reviewer 2 Report

Many thanks for the interesting study. Undoubtedly, the topic of vaccination is becoming very relevant, especially in light of the latest COVID-19 epidemic. In principle, I have no particular objections, except for a few points, within the framework of the discussion.

It seems possible to identify two potential factors that may influence vaccination processes.

And the first of them is the Covid epidemic. You have clearly indicated a number of aspects in your work and I would like to draw your attention to the following.

Vaccination against covid was not only positive for society. It showed the presence of a large number of people who do not trust vaccination and, moreover, in whose eyes the not always well-thought-out policy of universal vaccination once again convinced them of a conspiracy theory. Therefore, nearly a quarter of UK health workers were reluctant to receive regular SARS-CoV-2 vaccination [PMID: 36210437, PMID: 35033030]. This is aggravated by the fact that people call fear of becoming ill with the disease and being treated, and to allow social and family life to return to normal [PMID: 33713824] the main reasons for vaccination against covid. Therefore, children are given measles vaccination, which excludes these items from parents who make decisions for children.

Therefore, it would be interesting to analyze the motivations for different types of vaccination and find effective mechanisms (of course, in the context of future work).

The second aspect concerns the Russian-Ukrainian war and tensions in the Middle East, which brought a significant number of refugees to the United Kingdom, most of them children and women. And tolerance to vaccination in the former Soviet countries (Ukraine) is quite low (According to the latest available data, 85% of eligible children in Ukraine received their first dose of measles vaccine in 2020. Although this was a significant improvement compared to the low of 42% in 2016 and a major achievement for the country, WHO recommends a vaccine coverage of 95% or higher each year to achieve and maintain herd immunity and protect the population  https://www.who.int/europe/news/item/27-04-2022-ukraine--immediate-steps-needed-to-prevent-a-measles-outbreak-due-to-the-ongoing-war-and-low-vaccination-rates--warns-who . ).

Author Response

Please see the attachment: Response to Reviewer 2 MMR paper_Thompson_20 Jan 2023 final
